# BEHAVIOUR DISTILLATION

**Andrei Lupu[1,2], Chris Lu[1], Jarek Liesen[3], Robert Tjarko Lange[3] & Jakob Foerster[1]**
[1]University of Oxford [2]Meta AI [3]Technical University Berlin
Correspondence at `alupu@meta.com`

## ABSTRACT

Dataset distillation aims to condense large datasets into a small number of synthetic examples that can be used as drop-in replacements when training new models. It has applications to interpretability, neural architecture search, privacy, and continual learning. Despite strong successes in supervised domains, such methods have not yet been extended to reinforcement learning, where the lack of a fixed dataset renders most distillation methods unusable. Filling the gap, we formalize *behaviour distillation*, a setting that aims to discover and then condense the information required for training an expert policy into a synthetic dataset of state-action pairs, *without access to expert data*. We then introduce Hallucinating Datasets with Evolution Strategies (HaDES), a method for behaviour distillation that can discover datasets of *just four* state-action pairs which, under supervised learning, train agents to competitive performance levels in continuous control tasks. We show that these datasets generalize out of distribution to training policies with a wide range of architectures and hyperparameters. We also demonstrate application to a downstream task, namely training multi-task agents in a zero-shot fashion. Beyond behaviour distillation, HaDES provides significant improvements in neuroevolution for RL over previous approaches and achieves SoTA results on one standard supervised dataset distillation task. Finally, we show that visualizing the synthetic datasets can provide human-interpretable task insights.

## 1 INTRODUCTION

Dataset distillation (Wang et al., 2018) is the task of synthesizing a small number of datapoints that can replace training on a large real datasets for downstream tasks. Not only a scientific curiosity, distilled datasets have seen applications to core research endeavours such as interpretability, architecture search, privacy, and continual learning (Lei & Tao, 2023; Sachdeva & McAuley, 2023). Despite a series of successes on vision tasks, and more recently in graph learning (Jin et al., 2021) and recommender systems (Sachdeva et al., 2022), distillation methods have not yet been extended to reinforcement learning (RL). This is because they generally make strong assumptions about prior availability of an expert (or ground truth) dataset.

To address this gap in the literature, we introduce a new setting called *behaviour distillation*[1] that aims to discover and condense the information required for training an expert policy into a synthetic dataset of state-action pairs, *without access to an expert.* Unlike dataset distillation, which simply replaces a hard supervised learning task by an easier one, behaviour distillation solves two challenges at once: the *exploration problem* (discovering trajectories with high expected return) and the *representation learning problem* (learning to represent a policy that produces those trajectories), both of which are fundamental to deep reinforcement learning.

Thus, behaviour distillation aims to produce a dataset that obviates the need for exploration, essentially "pre-solving" the environment. As such, a behaviour distillation dataset does not encode a summary of the full environment, but only a summary of an expert policy in that environment. In other words, it reduces the joint problems of data collection (i.e. exploration) and sequential learning on a large amount of non-stationary data to one of *supervised* learning on a tiny amount of *stationary non-sequential synthetic* data, such as the example datasets in Fig. 1.

---

[1]The term was also recently used by Furuta et al. as a near-synonym for policy distillation.

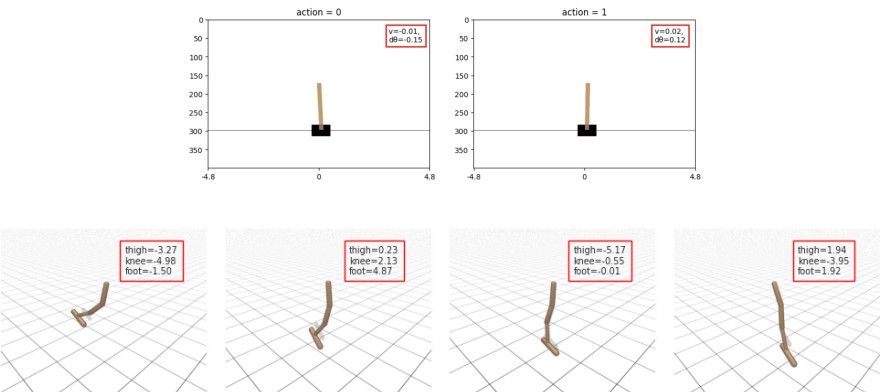

Figure 1: *Entire* synthetic datasets required to train an optimal *Cartpole* policy (top) and an expert *Hopper* policy with behaviour cloning (bottom). The state-action pairs help interpret the learned policies. Red box contains observation features for Cartpole and action labels (torques) for Hopper.

Motivated by the challenge of behaviour distillation, we introduce Hallucinating Datasets with Evolution Strategies (HaDES), a method based on a meta-evolutionary outer loop. Specifically, HaDES optimizes the *synthetic dataset* using a bi-level optimization structure, which uses evolutionary strategies (ES) to update the datasets on the outer loop and supervised learning ("behaviour cloning") on the current dataset in the inner loop. The *fitness function* for ES is the *performance* of the policy at the end of the supervised learning step. We show that the generated datasets can be used to retrain policies with vastly different architectures and hyperparameters from those used to produce the datasets and achieve competitive returns to training directly on the original environment while doing behaviour cloning on less than 1/10th or in some cases less than 1/100th of *a single episode* worth of data. We also demonstrate the applicability of these datasets to downstream tasks and open-source them in the hope of accelerating future research. Furthermore, while HaDES is tailored to behaviour distillation, we show it is also competitive when applied to popular computer vision dataset distillation benchmarks.

There is a recent resurgence of interest in evolutionary strategies (ES) for machine learning, fuelled by their generality and applicability to non-differentiable objectives, as well as to long-horizon tasks with delayed rewards Lu et al. (2023); Salimans et al. (2017). However, current evolutionary optimization methods are limited in the number of parameters that can be evolved, since a large number of parameters combined with a large population size induce a large memory footprint, as we show in Section 5.1. This limits the use of ES for training large neural networks.

To tackle this issue, we adapt HaDES into an alternative parametrization and training scheme for neuroevolution by not resampling the initial weights of the inner loop policy. This parametrization has the benefit of scaling independently of the number of parameters in the evolved policy, thereby reducing the memory footprint and resulting in competitive performance across multiple environments when compared to vanilla ES.

Our main contributions are:

1. We formalize the setting of *behaviour distillation*, which extends the principles of dataset distillation to reinforcement learning.(Section 4.1);

2. We introduce HaDES, the first method for behaviour distillation (Section 4);

3. We show that a minor change to our method provides a parametrization for neuroevolution through ES that reduces its memory (Section 5.1).

4. We demonstrate empirically that HaDES can produce effective synthetic datasets for challenging discrete and continuous control environments that generalize to training policies with a large range of architectures and hyperparameters (Section 5.2);

5. We use the synthetic datasets for a downstream task: quickly training a multi-task agent from datasets produced for individual environments (Section 5.3);

6. We achieve SoTA for a common dataset distillation benchmark with HaDES (Section 5.4);

7. We open-source our code and synthetic datasets under `https://github.com/FLAIROx/behaviour-distillation`.

## 2 RELATED WORKS

### 2.1 DATASET DISTILLATION

Efforts to reduce the amount of training data required for machine learning can be traced back to reduced support vector machines (Lee & Mangasarian, 2001; Lee & Huang, 2007; Wang et al., 2005). In deep learning, Bachem et al. (2017) and Coleman et al. (2019) have called *coreset selection* the problem of selecting a small number of representative examples from a dataset that suffice to train a model faster without degrading its final performance.

Wang et al. forgo this restriction to real examples in favour of producing synthetic datasets, coining the task of *dataset distillation*. Since then, numerous approaches have been proposed to distill supervised learning datasets (Lei & Tao, 2023). Most involve a bi-level optimization procedure and can be divided into four big categories (Sachdeva & McAuley, 2023). Gradient matching methods (Zhao et al., 2020) aim to minimize the difference in gradient updates that a model receives when training on the synthetic vs. the real dataset, while trajectory matching (Cazenavette et al., 2022) minimizes the distance between checkpoint parameters of models trained on either dataset. Unfortunately, neither of these techniques is applicable to reinforcement learning without prior access to an expert policy, its checkpoints or at the very least a dataset of expert trajectories. More recently, Zhao & Bilen and Wang et al. directly align the synthetic dataset distribution to match the real one. While highly effective, such an approach is also inapplicable to reinforcement learning due to the non-stationary and policy-specific data distribution. The oldest and most closely related approach to our method is meta-model matching (Wang et al., 2018; Nguyen et al., 2020; Loo et al., 2022), which involves fully training a model on the synthetic data in the inner loop, while updating the dataset in the outer loop to minimize the model's loss on the real data. These works either compute expensive meta-gradients using back-propagation through time (Wang et al. (2018); Deng & Russakovsky (2022), BPTT), or use a neural tangent kernel rather than a finite width neural network in the inner loop such that they can compute the classifier loss on the target dataset in closed form (Nguyen et al., 2020). While these methods could be applied to RL by choosing an appropriate loss (e.g. REINFORCE (Williams, 1992)), we instead replace meta-gradients by an evolutionary approach in the outer loop, making the cost of the outer loop updates independent of both the network size and the number of updates in the inner loop. This is important since we can use hundred of policy updates in the inner loop in practice, making the use of BPTT prohibitively expensive.

A few other works have extended dataset distillation beyond image classification to graphs (Jin et al., 2021; 2022) and recommender systems (Sachdeva et al., 2022), but to the best of our knowledge no previous work has broken away from assuming access to a pre-existing target dataset. As such, our work is the first to break the data-centric paradigm and introduces the first general distillation method applicable to distillation in reinforcement learning.

### 2.2 NEUROEVOLUTION AND INDIRECT ENCODINGS

Neuroevolution (Schwefel, 1977) has been shown to perform comparably to reinforcement learning on several benchmarks (Such et al., 2017; Salimans et al., 2017). Part of our work can be viewed as a form of indirect encoding (Stanley et al., 2019) for neuroevolution – an alternative parameterization for evolving neural network weights. Rather than evolve the parameters of a neural network directly, indirect encoding evolves a "genotype" in an alternative representation (which is usually compressed) that then maps to the parameters. A well-known example is HyperNEAT (Stanley et al., 2009), a precursor to HyperNetworks (Ha et al., 2016), which evolves a smaller neural network to generate the weights of a larger one. Indirect encoding is desirable because evolution strategies can scale poorly in the number of parameters (Hansen, 2016). Our work, instead of evolving a neural network, evolves a small dataset on which we train a larger neural network with supervised learning.

Other related work has also evolved other aspects of reinforcement learning training. For example, other works have evolved RL policy objectives (Lu et al., 2022; Co-Reyes et al., 2021; Houthooft et al., 2018; Jackson et al., 2024) and environment features (Lu et al., 2023). Most related to our work is Synthetic Environments (Ferreira et al., 2022), which evolve neural networks to replace an environment's state dynamics and rewards to speed up training. Instead of evolving transition and reward functions and training with RL, our work evolves supervised data for behavioural cloning (BC). This greatly aids the interpretability and simplifies the inner-loop.

## 3 BACKGROUND

The goal of this paper is to discover a behavioural dataset which, in combination with supervised learning, solves a Markov Decision Processes. We describe and formalize the conceptual basis of our paper below.

### 3.1 REINFORCEMENT LEARNING

A Markov Decision Process (MDP) is defined by a tuple $\langle \mathcal{S}, \mathcal{A}, \mathcal{P}, \mathcal{R}, \gamma \rangle$ in which $\mathcal{S}, \mathcal{A}, \mathcal{P}, \mathcal{R}, \gamma$ define the state space, action space, transition probability function (which maps from a state and action to a distribution over the next state), reward function (which maps from a state and action to a scalar return), and discount factor, respectively. At each step $t$, the agent observes a state $s_t$ and uses its policy $\pi_\theta$ (a function from states to actions parametrized by $\theta$) to select an action $a_t$. The environment then samples a new state $s_{t+1}$ according to the transition function $\mathcal{P}$ and a scalar reward $r_t$ according to the reward function $\mathcal{R}$. The objective in reinforcement learning is to discover a policy that maximizes the expected discounted sum of rewards:

$$J(\theta) = \mathbb{E}_{\pi_\theta} \left[ \sum_{t=0}^{\infty} \gamma^t r_t \right].$$ (1)

### 3.2 EVOLUTION STRATEGIES

Many reinforcement learning algorithms use the structure of the MDP to update the policy using gradient-based methods and techniques such as the Bellman equation (Bellman, 1966). An alternative approach is to treat the function $J(\theta)$ as a blackbox function and directly optimize $\theta$. One popular approach to this is known as Evolution Strategies Salimans et al. (2017). Given an arbitrary function $F(\phi)$, ES optimizes the following smoothed objective:

$$\mathbb{E}_{\epsilon \sim N(0,I)}[F(\phi + \sigma\epsilon)],$$

where $N(0, I)$ is the standard multivariate normal distribution and $\sigma$ is its standard deviation. We estimate the gradient of $F$ by sampling noise from $N(0, I)$ and evaluating $F$ at the resulting points. Specifically, the gradient is estimated by:

$$\nabla_\phi \mathbb{E}_{\epsilon \sim N(\mathbf{0},I_d)}[F(\phi + \sigma\epsilon)] = \mathbb{E}_{\epsilon \sim N(\mathbf{0},I_d)} \left[ \frac{\epsilon}{\sigma} F(\phi + \sigma\epsilon) \right].$$

We then apply this update to our parameters and repeat the process. When applied to meta-optimization, ES allows us to optimize functions that would otherwise require taking meta-gradients through hundreds or thousands of update steps, which is often intractable Metz et al. (2021).

### 3.3 DATASET DISTILLATION

In the context of supervised learning, dataset distillation is the task of generating or selecting a proxy dataset that allows training a model to a similar performance as training on the original dataset, often using several orders of magnitude fewer samples. Formally, we assume a dataset $\mathcal{D} = \{x_i, y_i\}_{i=1}^N$, of which $\mathcal{D}_{\text{train}} \subset \mathcal{D}$ is the training (sub)set, and a training algorithm $alg$. Also, let $f_{alg(\mathcal{D})} : x_i \mapsto y_i$ be the classifier obtained by training on $\mathcal{D}$ with $alg$. Then, the dataset distillation objective is to find a synthetic dataset $\mathcal{D}_\phi, |\mathcal{D}_\phi| << |\mathcal{D}_{\text{train}}|$, such that

$$\mathbb{E}_{x,y \sim \mathcal{D}} L(f_{alg(\mathcal{D}_\phi)}(x), y) \approx \mathbb{E}_{x,y \sim \mathcal{D}} L(f_{alg(\mathcal{D}_{\text{train}})}(x), y),$$ (2)

where $\phi$ indicates that the synthetic dataset can be parametrized and learned, rather than being sampled. In practice, $|\mathcal{D}_\phi|$ is often set to a fixed number of examples, e.g. $|\mathcal{D}_\phi| = n|Y|$, where $|Y|$ is the total number of discrete classes, or to be determined by fixed number of parameters, i.e. $|\phi| = n(|x| + |y|)$. The latter formulation, being more permissive, admits factorized representations of the data, e.g. by representing $|\mathcal{D}_\phi|$ with a generative neural network. While this is a promising avenue for future work, we focus exclusively on non-factorized distillation, which allows for better interpretability and tractability of the synthetic dataset.

## 4 PROBLEM SETTING AND METHOD

### 4.1 BEHAVIOUR DISTILLATION

We introduce *behaviour distillation* as a parallel to dataset distillation. Rather than optimizing a proxy dataset to minimize a loss in supervised learning, behaviour distillation optimizes a proxy dataset that maximizes the discounted return in reinforcement learning, after supervised learning. More formally, the aim of behavioural distillation is to find a dataset $\mathcal{D}_\phi \in (\mathcal{S} \times \mathcal{A})^N$, where $N$ is the number of points in the dataset, to solve the following bi-level optimization problem:

$$\max_{\mathcal{D}_\phi} \ J(\theta^*(\mathcal{D}_\phi)) \tag{3}$$

$$\text{s.t. } \theta^*(\mathcal{D}_\phi) = \arg\min_\theta L(\theta, \mathcal{D}_\phi), \tag{4}$$

where $\theta$ are the parameters of the neural network used in the policy $\pi_\theta$, $J$ is the discounted sum of returns defined in Eq. (1) and $L$ is any supervised learning loss.

Crucially, this setting does not assume access to an expert policy or dataset, for two reasons. Firstly, expert data may not always be available or may not be easily compressible, for instance if the expert is erratic or idiosyncratic. Secondly, standard imitation learning is plagued by cascading errors: if $\pi$ is an imitation learning policy that deviates from some expert $\pi_{\text{expert}}$ with probability $\leq \epsilon$, then it is likely to end up off-distribution, further increasing its error rate. As such, it generally incurs a regret $J(\pi_{\text{expert}}) - J(\pi) \geq T^2\epsilon$, where $T$ is the episode horizon (Ross & Bagnell, 2010). Given dataset distillation is lossy, applying it naively would result in a large $\epsilon$ and therefore a poor performance for $\pi$. Since we ultimately care about maximizing the expected discounted return rather than reproducing some specific expert behaviour, this is how we formalize behaviour distillation.

### 4.2 HADES

To tackle behaviour distillation, we introduce our method "Hallucinating Datasets with Evolution Strategies" (HaDES). HaDES optimizes the inner loop objective to obtain $\theta^*$ using gradient descent, and optimizes the outer loop objective (the return of $\pi_{\theta^*}$) using ES (Salimans et al., 2017).

In the inner loop, HaDES uses the cross-entropy loss for discrete action spaces, and the negative log likelihood loss for continuous action, albeit other losses could be substituted as well. We provide pseudocode in Algorithm 1.

#### 4.2.1 POLICY INITIALIZATION

We further specify two variants of our methods, which have distinct use cases. They differ only in the way inner loop policies are initialized, which leads to different inductive biases. The variants described here are visualized in Fig. 2

The first variant is HaDES with *fixed* policy initialization, or HaDES-F. In this variant, we sample a single policy initialization $\theta_0$ at the very beginning of meta-training. The policy is re-trained every inner-loop, but always starting from this fixed $\theta_0$.

The second variant is HaDES with *randomized* policy initialization, or HaDES-R. In this variant, we use multiple ($k \geq 2$) policy initializations $(\theta_0^1, ..., \theta_0^k)_i$ in the inner loop, and we resample the initializations randomly at every generation $i$.

HaDES-F has a stronger inductive bias in that it is only optimizing $\mathcal{D}_\phi$ for a single initialization. We expect it might be able to "overfit" on that initialization and achieve higher policy returns, but at the cost of the synthetic dataset having poor generalization properties to other initializations. HaDES-F will therefore be stronger for neuroevolution (where only the final return of the specific policy matters), but weaker for behaviour distillation.

HaDES-R has a weaker inductive bias in that it optimizes $\mathcal{D}_\phi$ for a range of initializations. We expect this to result in decreased policy returns, but the synthetic dataset to be a useful artifact of training that generalizes to unseen initializations or even unseen policy architectures. HaDES-R will therefore be a better choice for behaviour distillation, but a weaker one for neuroevolution. We confirm both those intuitions empirically in Section 5.

Figure 2: Left: Standard neuroevolution. Middle: HaDES-F. Right: HaDES-R. HaDES-F uses a single fixed policy initialization. HaDES-R samples $k \geq 2$ policy initializations every generation.

## 5 EXPERIMENTS

Next, we show that HaDES can be successfully applied to a broad range of settings. More specifically, we investigate the following four applications:

1. We demonstrate the effectiveness of HaDES as an indirect encoding for neuroevolution in RL tasks (Section 5.1), with an analysis of the distillation budget in Appendix C.1.

2. We show that synthetic datasets successfully generalize across a broad range of architectures and hyperparameters (Section 5.2).

3. We show that the synthetic datasets can be used to train multi-task models (Section 5.3).

4. We show that while our focus is on RL, our method is also competitive when applied to dataset distillation in supervised settings (Section 5.4).

We refer the reader to Appendix B.2 for experimental details, including runtime comparisons.

### 5.1 PERFORMANCE EVALUATION OF HaDES-DISCOVERED RL DATASETS

We first test the effectiveness of HaDES as a way of training competitive policies in 8 continuous control environments from the Brax suite. Fig. 3a shows the performance of HaDES with a fixed policy initialization (HaDES-F), HaDES with two randomized initializations (HaDES-R), and direct neuroevolution through ES. HaDES-F achieves the highest return across the board, while HaDES-R also matches or beats the baseline in $6/8$ environments. In Humanoid-Standup, our method discovers a glitch in the collision physics, propelling itself into the air to achieve extremely high returns[2].

In MinAtar, we find that HaDES-F outperforms the ES baseline in Breakout, and matches it in SpaceInvaders and Freeway. We hypothesize MinAtar is a harder setting for our method due to the symbolic (rather than continuous) nature of the environment. For both settings, we also plot the performance of PPO policies after $5 \times 10^7$ training steps. While there is still a performance discrepancy between ES and RL, HaDES narrows the gap significantly.

We use policy networks with width 512 and a population size of 2048 for HaDES, but are forced to cut the network widths by half for the ES baseline on MinAtar due to memory constraints. Indeed, ES requires the entire population to be allocated on a single device at once when a new population is generated and when estimating the meta-gradient at each generation. While distributed approaches to the outer loop computation are feasible, they involve an engineering overhead, which our method alleviates. As such, our method is drastically more memory efficient, enabling larger populations and network sizes with minimal code changes. We also run HaDES with width 256 for completeness.

### 5.2 HaDES DATASETS GENERALIZE ACROSS ARCHITECTURES & HYPERPARAMETERS

We now turn our attention to the datasets themselves, and use them to train new policies in a zero-shot fashion, i.e. without additional environment interactions. We take two synthetic datasets for Hopper – one generated with HaDES-R and another with HaDES-F – and use them to train new policies from scratch. For each dataset, and for each of 7 different policy network sizes, we train 50 policies, each using a different learning rate and number of training epochs. In particular, the learning rates span 3 orders of magnitude and training epochs ranged uniformly between 100 and

---

[2]video link

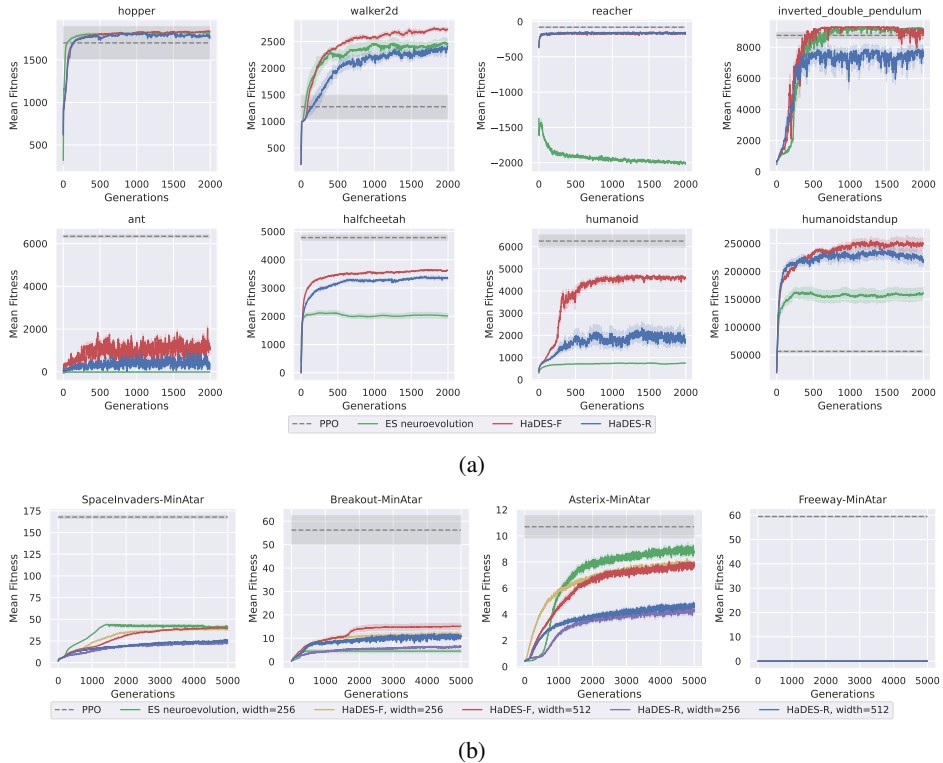

(a)

(b)

Figure 3: HaDES trains competitive policies on a) Brax, using 64 state-action pairs and b) MinAtar using 16 state-action pairs. For each environment, we show the mean return of the population at each generation for HaDES-F, HaDES-R and direct neuroevolution through ES, as well as the PPO final performance after $5 \times 10^7$ steps. HaDES-F matches or outperforms direct ES on all Brax environments, outperforms ES in one out of four MinAtar environments and matches it in two others. We also observe a significant gap between HaDES-F and HaDES-R, as predicted in Section 4.2.

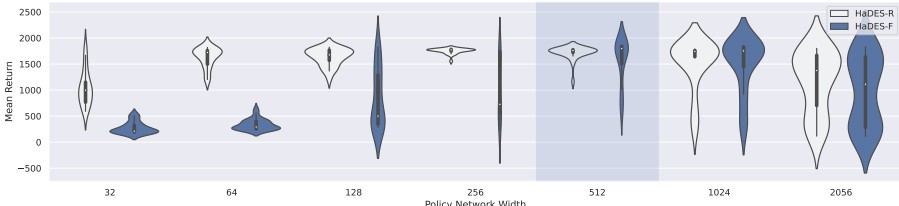

Figure 4: Hopper dataset transfer to other architectures and training parameters. We take a synthetic dataset of 64 state-action pairs evolved for policy networks of size **512** (highlighted) and use it to train policies with varying widths and 50 hyperparameter combinations per width. We plot the top 50% within each width group. HaDES-F indicates that the dataset was trained with a fixed $\pi_0$. The HaDES-R dataset was trained with randomized $(\pi_0^1, ..., \pi_0^k)_i$ and generalizes much better across all architectures and training parameters. This holds generally across environments (see Appendix C.2).

500. For each dataset and width, we discard the worst 25 policies and plot the return distribution of the remaining 25 in Fig. 4.

The datasets were evolved for policies of width 512, a fixed learning rate, and a fixed number of epochs, but readily generalize out of distribution to training with different settings and architectures. In particular, we see that the HaDES-R dataset is more robust to changes in both policy architecture and training parameters than the HaDES-F dataset, which incorporates a stronger inductive bias.

We hypothesize that the generalization properties of the synthetic datasets can further be improved by randomizing not only the policy initialization, as in HaDES-R, but also architectures and training

parameters, further reducing some of the inductive biases present in our implementation. How to best navigate the trade-off between generalization, dataset size and policy performance remains an interesting question for future work.

## 5.3 HaDES Datasets Can Be Applied to Zero-shot multi-tasking

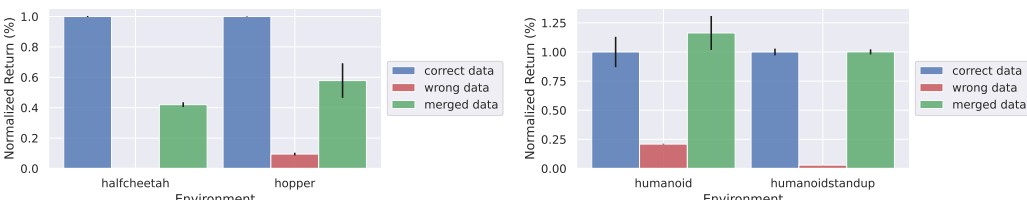

Figure 5: We use the synthetic datasets to train multi-task agents without any additional environment interaction. We plot the normalized fitness of agents trained either on the *correct* dataset for their environment, the *wrong* dataset for their environment, or a combined dataset, merged through concatenation and zero-padding to have the observation sizes match. Left: we train multi-task agents that achieve $\sim 50\%$ normalized fitness for Halfcheetah and Hopper. Right: we train agents that achieve $\gtrsim 100\%$ normalized fitness for Humanoid and Humanoidstandup. The multi-task policy architecture and training parameters were not optimized. We plot mean $\pm$ stderr. across 10 seeds. This shows that synthetic datasets can accelerate future research on RL foundation models.

We showcase an application of the distilled dataset by training multi-task Brax agents in a zero-shot fashion. To this end, we merge datasets for two different environments in the following way. Let $\mathcal{D}_1 = (\mathcal{S}_1, \mathcal{A}_1)$ and $\mathcal{D}_2 = (\mathcal{S}_2, \mathcal{A}_2)$ be the two datasets, each with 64 state-action pairs. We first zero pad the states in $\mathcal{S}_1$ to the left **and** zero pad the states in $\mathcal{S}_1$ to the right, such that each padded state has now the combined dimension $|s_1| + |s_2|$, where $\mathcal{S}_1 \subset \mathbb{R}^{|s_1|}$ and $\mathcal{S}_2 \subset \mathbb{R}^{|s_2|}$. We then do the same for the actions, such that each padded action now has size $|a_1| + |a_2|$, where $\mathcal{A}_1 \subset \mathbb{R}^{|a_1|}$ and $\mathcal{A}_2 \subset \mathbb{R}^{|a_2|}$. Finally, we take the union of both padded datasets to build a merged dataset $\mathcal{D}_{\text{merged}}$ with 128 state-action pairs. We then train agents on either $\mathcal{D}_1$, $\mathcal{D}_2$ or $\mathcal{D}_{\text{merged}}$ using behaviour cloning and report the normalized performance on both environments in Fig. 5.

We perform this experiment for two pairs of environments: (Halfcheetah, Hopper) and (Humanoid, Humanoid Standup). Blue indicates the baseline performance of training on $\mathcal{D}_i$ and evaluating on environment $i$. Red shows the performance of training on $\mathcal{D}_i$ and evaluating on environment $-i$, and is a loose proxy for how much the data from one environment helps to learn a policy for the other. Finally, green shows the performance of policies trained on $\mathcal{D}_{\text{merged}}$. We see that the multi-task agents achieve roughly 50% of the single-task performance in the first pair of environments, but see no loss in performance in the second pair.

This shows that the synthetic datasets evolved by HaDES can vastly accelerate future research on RL foundation models. Indeed, given those datasets, training new models takes only a few seconds, and makes it possible to experiment with architectures and multi-task representation learning at a fraction of the original computational cost. Furthermore, it allows studying the properties of cross-task parameter sharing and representation learning in isolation, separately from the exploration issues inherent to reinforcement learning.

## 5.4 HaDES Can Be Applied to Supervised Dataset Distillation

While the focus is on behaviour distillation in RL, our method is readily applicable to the standard dataset distillation setting by replacing environment return with a cross-entropy loss on some target dataset. We apply our method to dataset distillation with 1 image/class in MNIST and FashionMNIST, with results in Table 1. We use the cross-entropy loss on the training set as the fitness for HaDES and report the mean accuracy on the test set obtained by training classifiers on the final synthetic dataset. We run 3 different seeds and train 20 classifiers for each final datasets. Similar to Zhao et al. (2020) and Zhao & Bilen (2021), we report the mean and standard deviation across all 60 final classifiers and compare against the best method in these settings, namely RFAD Loo et al.

|  | MNIST | | FashionMNIST | |
|---|---|---|---|---|
|  | RFAD | Ours | RFAD | Ours |
| 1 img/cls | **94.4 ± 1.5** | 90.1 ± 0.3 | 78.6 ± 1.3 | **80.2 ± 0.4** |

Table 1: Test set accuracy of classifiers trained on datasets composed of 1 image per class. We compare to RFAD, which is the SotA for non-factorized dataset distillation on these datasets (Sachdeva & McAuley, 2023). RFAD uses a ConvNet architecture for testing, while we use a smaller CNN. Despite being designed for RL, our method is also competitive for image-based dataset distillation and achieves state-of-the-art distillation for 10-image FashionMNIST.

(2022). We find that HaDES performs competitively in 10 image MNIST and achieves state-of-the-art results in FashionMNIST. However, we were unable to scale our methods to CIFAR-10, which have many more parameters due to being RGB rather than greyscale.

In tuning our method, we find that the two single most important parameters are the outer learning rate and the dataset initialization. We find that initializing the dataset at the class-wise mean of the data works best. This is similar to findings by Zhao & Bilen, who also find that warm starting the dataset performs better than initializing from scratch.

### 5.5 HaDES-explainability

The final benefit of the HaDES datasets is that the synthetic examples lend themselves to interpretability. Going back to Fig. 1, we see that the resulting datasets have intuitive properties. For instance, the two state dataset for Cartpole captures that the policy should go left if the pole is leaning left and go right otherwise.

Performing an explainable RL study lies beyond the scope of this paper, but in a critical analysis, Atrey et al. highlight the importance of taking a hypothesis-driven approach to explaining deep RL policies, formulating possible explanations, then testing them rigorously with ablations and careful experiments. With that in mind, we argue that our synthetic datasets are an effective starting point for such hypothesis-testing, for instance by applying transformations to datasets and observing how it impacts the trained policies.

## 6 Discussion and Conclusion

In this paper, we introduced a new parametrization for policy neuroevolution by evolving a small synthetic dataset and training a policy on it through behavioural cloning. We showed that our method produces policies with competitive return in continuous control and discrete tasks. Our method can be used for behaviour distillation, summarizing all relevant information about optimal behaviour in an environment into a small synthetic dataset. This "behaviour floppy disk" can quickly train new policies parametrized by a range of different architectures. We then demonstrated the utility of the distilled dataset by training multi-task models. We finished by showing that although our focus is on RL, our method also applies to vanilla dataset distillation in supervised learning, where we achieved state-of-the-art in one settings.

The main limitation of this work is of computational nature, since evolutionary methods require a large population to be effective. Furthermore, while our alternative parameterization enables us to evolve larger neural networks than standard neuroevolution, the number of parameters still grows linearly with the number of datapoints, especially in pixel-based environments which tend to be very highly dimensional. Tackling this issue, for instance by employing factorized distillation, is therefore a promising avenue for future work. Another downside of our work is the number of hyperparameters, since we need to tune both the ES parameters in the outer loop and the supervised learning ones in the inner loop. However, anecdotal evidence seems to indicate that ES can adapt to the inner loop parameters, for instance by increasing the magnitude of the dataset if the learning rate is low. Understanding the interplay between parameters would allow for faster an better tuning. A related approach would also be to evolve the inner loop parameters along with the dataset. Finally, possible applications of the distilled datasets are ripe for investigation, for instance in continual or life-long learning, and regularizing datasets to further promote interpretability.

REPRODUCIBILITY STATEMENT

To encourage reproducibility, we described the method in detail in Section 4.2, include pseudocode (Algorithm 1) and provide hyperparameters in Appendix B.1, in a format that corresponds directly to the configs used by our code. We also open-source our code and our synthetic datasets at `https://github.com/FLAIROx/behaviour-distillation`.

ACKNOWLEDGMENTS

Andrei Lupu was partially funded by a *Fonds de recherche du Québec* doctoral training scholarship.

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

## A HaDES Algorithm

---

**Algorithm 1** HaDES

---

**Require:** dataset size $n$, environment Env
**Require:** number of meta steps $T$, pop. size $P$, learning rate $\alpha$, std $\sigma$
1: Initialize $\mathcal{D}_\phi = \{(s,a)_1, \ldots, (s,a)_n\}$                  ▷ e.g. randomly or sampled
2: **for** meta_step $= 0, \ldots, T$ **do**
3:      Initialize $\xi \sim \mathcal{N}(0, \sigma)$
4:      **for** $i = 0, \ldots, P$ **do**
5:          **if** $i$ is even **then**
6:              Perturb $\mathcal{D}_\phi$ with noise $\xi_i = \xi$ to get $\mathcal{D}_i$           ▷ antithetic noise
7:          **else**
8:              Perturb $\mathcal{D}_\phi$ with noise $\xi_i = -\xi$ to get $\mathcal{D}_i$
9:              Update $\xi \sim \mathcal{N}(0, \sigma)$
10:          **end if**
11:          Initialize policy $\pi_\theta$
12:          Train policy $\pi_\theta$ on $\mathcal{D}_i$ using BC
13:          Unroll $\pi_\theta$ and compute expected return $J_i = J(\pi_\theta, \text{Env}|\mathcal{D}_i)$
14:      **end for**
15:      Approximate $\nabla_\phi J \approx \frac{1}{P\sigma} \sum_i J_i \xi_i$
16:      Update $\mathcal{D}_\phi = \mathcal{D}_\phi + \alpha \nabla_\phi J$
17: **end for**
18: Train policy $\pi_\theta$ on final $\mathcal{D}_\phi$
19: **return** $(\pi_\theta, \mathcal{D}_\phi)$

---

## B Implementation details

### B.1 Hyperparameters

| Parameter | Value |
|---|---|
| LR | 0.005 |
| NUM_ENVS | 4 |
| NUM_STEPS | 1024 |
| UPDATE_EPOCHS | 400 |
| MAX_GRAD_NORM | 0.5 |
| ACTIVATION | tanh |
| WIDTH | 512 |
| ANNEAL_LR | False |
| GREEDY_ACT | False |
| CONST_NORMALIZE_OBS | False |
| NORMALIZE_OBS | True |
| NORMALIZE_REWARD | True |
| popsize | 2048 |
| dataset_size | 64 |
| rollouts_per_candidate | 1 |
| n_generations | 2000 |
| sigma_init | 0.03 |
| sigma_decay | 1.0 |
| lrate_init | 0.05 |
| Evo. strategy | OpenES |

Table 2: Hyperparameters for HaDES in Brax. Top: inner loop parameters. Bottom: Outer loop parameters.

| Parameter | Value |
|---|---|
| NET | mlp |
| LR | 0.03 |
| NUM_ENVS | 8 |
| NUM_STEPS | 1024 |
| UPDATE_EPOCHS | 64 |
| MAX_GRAD_NORM | 0.5 |
| ACTIVATION | relu |
| WIDTH | 512 or 256 |
| ANNEAL_LR | True |
| GREEDY_ACT | False |
| CONST_NORMALIZE_OBS | True |
| NORMALIZE_OBS | False |
| NORMALIZE_REWARD | False |
| popsize | 2048 |
| dataset_size | 16 |
| rollouts_per_candidate | 2 |
| n_generations | 5000 |
| sigma_init | 0.5 |
| sigma_limit | 0.01 |
| sigma_decay | 1.0 |
| lrate_init | 0.05 |
| lrate_decay | 1.0 |
| Evo. strategy | SNES |
| temperature | 20.0 |

Table 3: Hyperparameters for HaDES in MinAtar. Top: inner loop parameters. Bottom: Outer loop parameters.

## B.2 EXPERIMENTAL DETAILS

For all RL tasks we use Brax (Freeman et al., 2021), a suite of continuous control environments, and MinAtar (Young & Tian, 2019), a set of Atari-like environments.

For dataset distillation, we report results on two image classification tasks: MNIST (LeCun, 1998), which is composed of handwritten digits, and FashionMNIST (Xiao et al., 2017), which features different clothing items.

For the evolutionary algorithm, we use OpenES Salimans et al. (2017) for Brax and image classification, and use SNES Wierstra et al. (2014) for MinAtar. In the inner loop, we minimize either the cross-entropy loss (discrete cases) or the negative log likelihood of the synthetic actions (continuous action cases). All of our runs use 8 Nvidia V100 GPUs and take between 1 and 17 seconds per outer loop generation. Detailed generation times are reported in Table 4. These times include outer loop operations (all methods), inner loop policy training (HaDES only), and inner loop policy evaluation (all methods). HaDES-R is slightly slower than HaDES-F since it trains two policies instead of just one. Because we train policies from scratch every generation, the times reported are strict upper bounds to how long it takes to train a policy on the final distilled datasets.

In image classification and MinAtar, we assign labels (i.e. classes or discrete actions) uniformly, whereas in Brax we evolve the dataset labels alongside the observations since the environments feature continuous actions.

We implement our algorithm in JAX (Bradbury et al., 2018) using the PureJaxRL (Lu et al., 2022), gymnax (Lange, 2022) and evosax (Lange, 2023) libraries to enable parallel training on hardware accelerators. We also use virtual batch normalization (Salimans et al., 2016) to stabilize training, which was previously found to be crucial in stabilizing ES (Salimans et al., 2017).

| Environment Name | ES Neuroevolution | HaDES-F | HaDES-R |
|---|---|---|---|
| Hopper | 4.0 | 5.6 | 6.3 |
| Walker2d | 3.4 | 7.9 | 8.5 |
| Reacher | 2.7 | 6.7 | 7.2 |
| Inverted Double Pendulum | 2.6 | 4.4 | 4.6 |
| Ant | 6.6 | 11.1 | 12.1 |
| Halfcheetah | 10.7 | 14.8 | 16.9 |
| Humanoid | 7.8 | 13.9 | 15.1 |
| HumanoidStandup | 8.6 | 14.8 | 16.4 |
| SpaceInvaders-MinAtar | 1.6 | 1.5 | 1.6 |
| Breakout-MinAtar | 1.3 | 1.6 | 1.8 |
| Asterix-MinAtar | 1.9 | 2.1 | 2.4 |
| Freeway-MinAtar | 2.3 | 2.2 | 2.9 |

Table 4: Runtime of the different neuroevolution methods in seconds/generation. Times averaged over 3 seeds rounded to the nearest tenth of a second. Standard deviation omitted, but the difference between the fastest and slowest runs for any setting is usually smaller than 0.2 seconds.

## C  ADDITIONAL RESULTS

### C.1  IMPACT OF DISTILLATION BUDGET ON PERFORMANCE

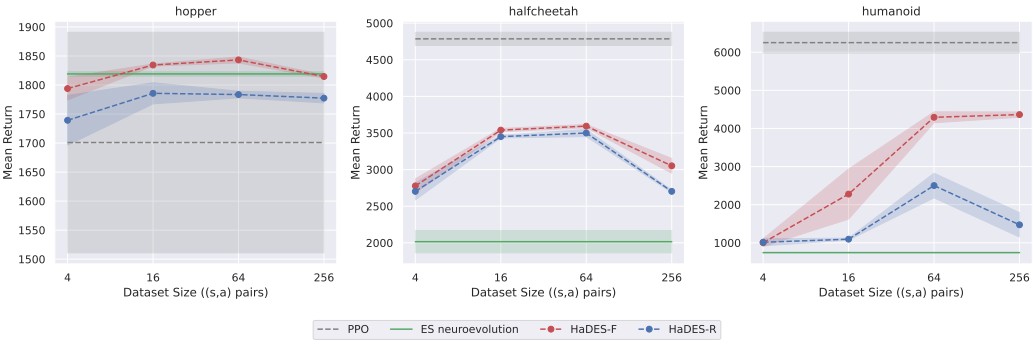

Figure 6: Final return of HaDES policies as a function of distillation budget (i.e. dataset size). ES neuroevolution and PPO returns also plotted for reference.

In Fig. 6, we investigate the impact of the distillation budget on the final performance of policies trained with HaDES on three different environments. What we observe is that dataset sizes that are too small degrade performance, likely because they cannot contain all the information required to train an expert policy. This is particularly noticeable in Humanoid, where for a dataset of 4 state-action pairs, the score drops as low as 1013. However, a score of 1000 corresponds to a humanoid policy that keeps its balance and stays immobile, with lower scores indicating that the policy falls and causes an early termination. This is an indication that for distillation budgets that are too low to capture expert behaviour, HaDES does not fail to learn, and will still optimize return within the constraints of the budget.

On the opposite end of the spectrum, we also observe return dropping for large dataset sizes ($|\mathcal{D}| = 256$), despite the increased expressivity. This is possibly due to ES (and therefore HaDES) scaling poorly to a large number of parameters. This problem can be alleviated by relying on better ES methods, or by using a factorized approach to distillation.

### C.2  DATASET GENERALIZATION ACROSS ARCHITECTURES AND HYPERPARAMETERS

Here we plot generalization plots for additional environments. As expected, HaDES-R generalizes better than HaDES-F both to new hyperparameters and to new architectures.

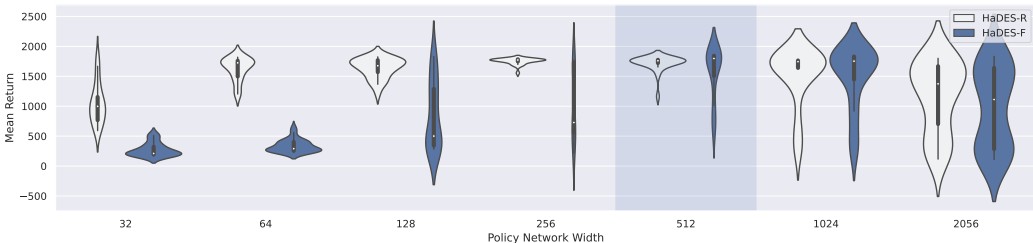

Figure 7: Dataset transfer to hopper architecture and training parameters.

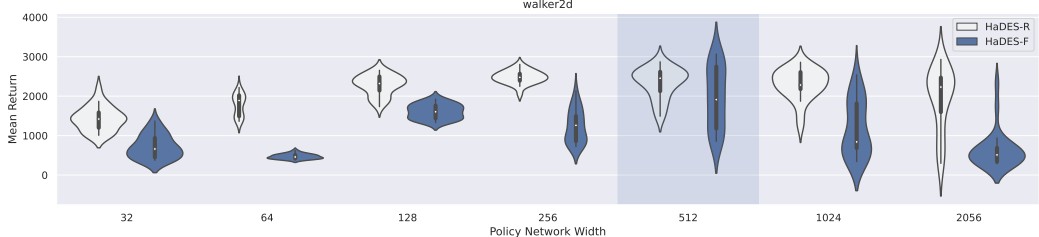

Figure 8: Dataset transfer to walker2d architecture and training parameters.

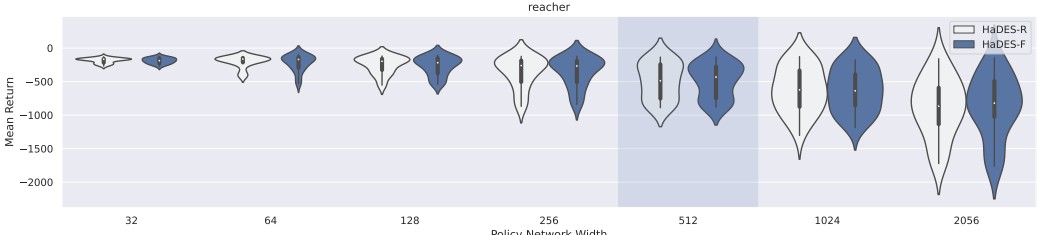

Figure 9: Dataset transfer to reacher architecture and training parameters.

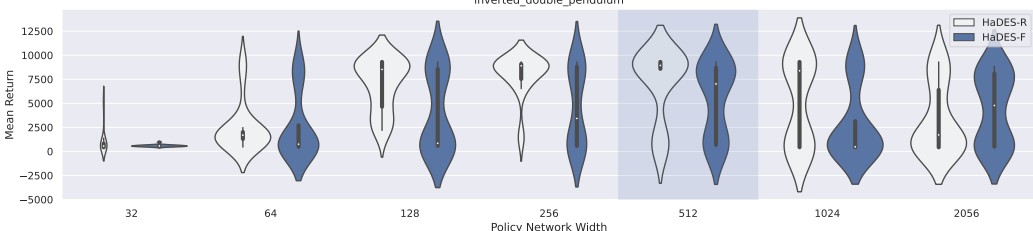

Figure 10: Dataset transfer to inverted_double_pendulum architecture and training parameters.

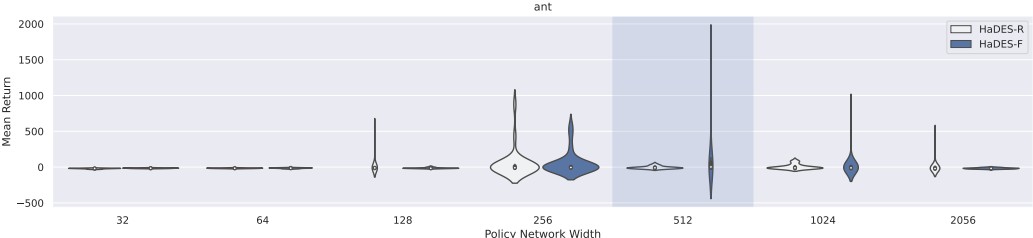

Figure 11: Dataset transfer to ant architecture and training parameters.

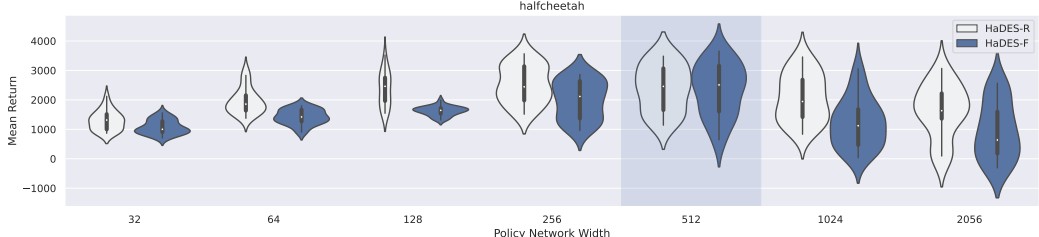

Figure 12: Dataset transfer to halfcheetah architecture and training parameters.

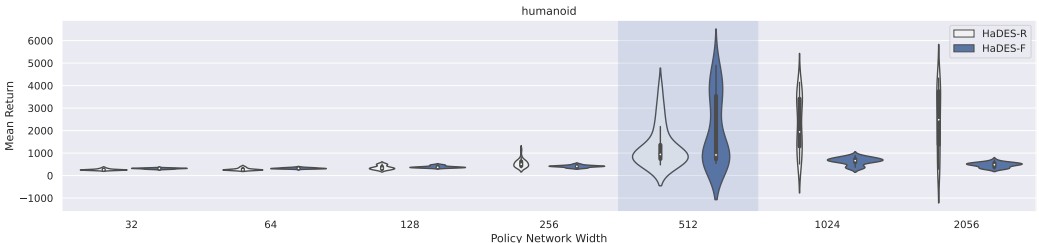

Figure 13: Dataset transfer to humanoid architecture and training parameters.

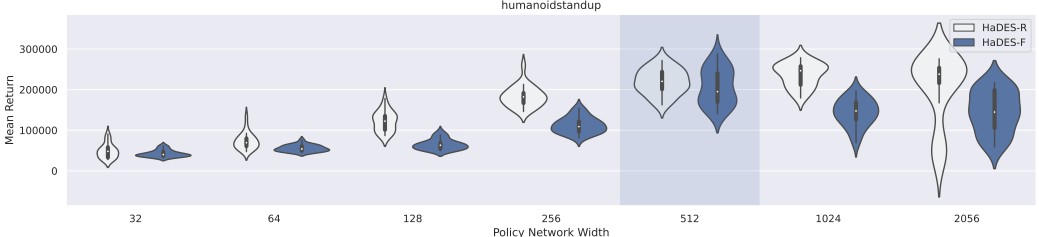

Figure 14: Dataset transfer to humanoidstandup architecture and training parameters.

