# OpenReview forum: "Behaviour Distillation"
_ICLR.cc/2024/Conference — ICLR 2024 poster_

### Official Review · Reviewer_ZN7t · 2023-10-27

**Soundness:** 3 good
**Presentation:** 3 good
**Contribution:** 3 good
**Rating:** 8
**Confidence:** 2

**Summary:**

The authors attempt to transfer the concept of dataset distillation to the realm of reinforcement learning. To that end, they introduce the concept of behaviour distillation, which is trying to condense a dataset of state-action pairs required for training a RL agent. After introducing this concept, they propose a new method named HADES, that implements this concept and is able to distill the initial state-actions pairs into a dataset of just four state-action pair that, when trained upon by an agent can make it reach competitive performance.

**Strengths:**

- Transfer the concept of dataset distillation to reinforcement learning
- The condensed dataset can result in a quite small one
- The results on supervised classification dataset distillation also seem good

**Weaknesses:**

- 1) I think the neuroevolution part is quite distinct from the dataset distillation part. It should be explained clearly why you need to use a neuroevolution technique instead of a more classical technique
- 2) Maybe the naming is a bit confusing, if the method proposed is meant to distill the dataset only (and not train on it later), then it would be clearer to name the experiments as Method + HADES (whenever training a method on top of hallucinated dataset).
- 3) If I understood correctly, one advantage of HADES is that it can distill behaviour by looking at some subpart of the episode (like 1/10th), and condense the gathered state-action pairs into much smaller dataset. However, from the rest of the paper, it's not clear whether they are some computational advantages of training on top of these condensed dataset. They should be made clearer. Indeed, by looking at the RL plots, in most of the cases, it's not clear if the red curve trains way faster than the green curve (apart from a few environments like humanoid and breakout-miniAtari

**Questions:**

- 1) If I understand correctly, HADES first distills a dataset into a condensed state-action pair and then trains a classical policy on it (as in classical dataset distillation). What I don't understand is the link with the neuroevolution part. Is this part requires only for distillation or is it also required for training on top of the distilled dataset ? In the latter case, it would be interesting to see how other methods that do not use neuroevolution perform when trained on the condensed dataset.
- 2) In the results plot of reinforcement learning, does the number of generation directly correlates to the training time ? Or are some generations quicker to run for methods trained on condensed dataset (that implies the inner loop would be faster I guess) ? I think the computational advantage of training on top of condensed dataset (if any) should be made clearer

---

> ### Author Response · Authors · 2023-11-17
> **Response to R3**
>
> We thank reviewer ZN7t for their comments.
>
> 0. > _"HaDES [...] is able to distill the initial state-actions pairs into a dataset of just four state-action pair"_
>
> We formulate Behaviour Distillation without access to expert data, as mentioned in the abstract, introduction and section 4.1. This means there are no "initial state-action pairs". Also, the size $N$ of the distilled dataset is a hyperparameter we choose. Figures 1 shows datasets with $N=2$ and $N=4$, and in section 5 we report results with $N=64$ and $N=16$.
>
> **Response to Weaknesses:**
>
> 1. > _"explain [...] why you need to use a neuroevolution technique instead of a more classical technique"_
>
> Neuroevolution covers all methods optimising a neural network using Evolutionary Strategies (ES) [1]. We do not use neuroevolution. HaDES, and in particular HaDES-F, can act as a neuroevolution method, as we explain in section 4.2 and demonstrate in section 5.1.  We use ES instead of a gradient-based approach such as Backpropagation Through Time (BPTT) because BPTT is expensive when training the inner loop policy for many steps. For context, we use 400 inner loop training steps, while [3] only uses 10 to 30.  We have now clarified this in section 2.1.
>
> 2. > "...it would be clearer to name the experiments as Method + HADES"
>
> HaDES is a method that takes in an environment M and outputs a tuple $(D_\phi, \pi_{\theta^*})$, where $D_\phi$ and $\pi_{\theta^*}$ are the solutions to the optimization problem in eq. 3 and 4. HaDES uses ES in the outer loop and behaviour cloning (supervised learning) in the inner loop. Training a policy on the synthetic dataset is integral to HaDES, so we cannot split the name into "method + HaDES".
>
> 3. > "looking at some subpart of the episode (like 1/10th)..."
>
> HaDES does not divide episodes at all. It evolves the synthetic datasets from scratch. In section 1, we say that the final dataset contains "1/10th [...] of a single episode **worth** of data." This is about the final magnitude of the dataset, not about how HaDES works.
>
> > "... computational advantages of training on top of these condensed dataset."
>
> The usual episode is 1000 transitions, and our datasets are at most 64 states (<10% of an episode). In contrast, training a policy using RL would require ~50k episodes [2]. Training on the distilled datasets uses between **6 and 8 orders of magnitude less data** that standard RL training, making training policies extremely fast.
>
> We have also updated Section B.2 of the manuscript with details of the GPU time required to run each method.
>
> > "looking at the RL plots [...] it's not clear if the red curve trains way faster than the green curve"
>
> We are not sure what the reviewer is trying to say. HaDES-F (red curve) reaches statistically higher return than Vanilla ES (green curve) in 7/12 environments reported, reaches similar returns in 4/12 environments and reaches statistically lower returns in 1/12 environments. If the reviewer could please clarify what they mean, we would be happy to respond further.
>
> **Answer to Questions:**
>
> 1. > "HADES first distills a dataset into a condensed state-action pair and then trains a classical policy on it (as in classical dataset distillation)."
>
> That is incorrect. As stated previously, HaDES does not take a dataset as input and does not perform classical dataset distillation. We have also expanded section 4.1 with details on why we do not rely on an expert dataset.
>
>  > "...neuroevolution part. Is this part requires only for distillation or is it also required for training on top of the distilled dataset ? In the latter case, it would be interesting to see how other methods that do not use neuroevolution perform when trained on the condensed dataset."
>
> Hopefully, our explanation under W1 clarified that we do not use neuroevolution. We use ES (which is different from neuroevolution) to optimise the synthetic dataset during the distillation process. When we train the policy (in the inner loop or on the final distilled dataset), we simply use behaviour cloning (i.e. supervised learning).
>
> 2. > "In the results plot of reinforcement learning, does the number of generation directly correlates to the training time ?"
>
> For each method and each environment, the time/generation is constant. Generally, HaDES takes a bit longer per generation since it trains a policy every inner loop, as opposed to ES, which must only evaluate the policy. We have added a time/generation breakdown to section B.2 of the appendix.
>
> We hope our response has clarified any misunderstanding and that we have addressed the reviewer's feedback. If so, we would appreciate if they would consider increasing their score.
>
> [1] https://link.springer.com/referenceworkentry/10.1007/978-0-387-30164-8_589
>
> [2] ReLU to the rescue, https://arxiv.org/abs/2306.01460
>
> [3] Dataset Distillation, https://arxiv.org/pdf/1811.10959.pdf
>
> [4] Efficient Reductions for Imitation Learning, https://proceedings.mlr.press/v9/ross10a.html

---

> > ### Comment · Reviewer_ZN7t · 2023-11-20
> >
> > Thanks to the authors for clarifying some points and answering my questions. Below I ask a few more to be sure that I understood the paper correctly in order to give a better review.
> >
> > -  You mention that for this method, the dataset distillation cannot be combined with another method (cf. answer to the question on why no naming Method + HADES dataset). I think I understood your answer in that once you have a distilled dataset, learning a policy on top of it is similar to "offline" supervised training and thus you cannot really use another kind of policy learning algorithm for that task. Is that correct ?
> >
> > - I understand now why it is necessary to use evolutionary strategies to avoid unfeasible computation of meta-gradients on a big number of inner steps. This is a good idea. As I am not familiar with these strategies, I am wondering however how well they can work when the state space gets bigger (in which case approximating the gradient with noise seems a more difficult problem). What is the typical "biggest" state space size that you have considered in your experiments ? (Could it for instance work on a state that is compounded of the features extracted from a resnet, of size 512).
> >
> > - In my previous questions, I was referring incorrectly sometimes to "dataset" and saying the HADES algorithm requires to access a "dataset" because I thought you were collecting a buffer of state-action pairs that you would then sample from as done in some algorithms. From your answer I understood that it is not the case, you learn this dataset online from episodes, is that correct ?
> >
> > - About the comment on training speed, you can ignore it, I thought that these curves were comparing a method that is training on normal episodes vs methods that are training on distilled dataset and I was surprised to see no speed improvement. As I understand now, HADES is learning both a dataset and a policy, so these curves are just comparing HADES to ES and it makes sense that there is no speed improvement for the dataset learning part. I understand now that only Figure 4 looks at the results of training on top of HADES dataset.
> >
> > - After all of this, I think that the Figure 4 is the most important figure since it really shows the capacity of the distilled dataset. In my opinion it would be better to highlight this and maybe show it for different kind of environments instead of just Hopper. The fact that the policy learned by HADES gets better returns is less interesting in my opinion (and then it should be compared to more policy-learning algorithms additionally to ES).
> >
> > I would be happy to increase my score, but I would like to be sure that I understand the manuscript correctly.

---

> > > ### Author Response · Authors · 2023-11-20
> > > **2nd Response to R3**
> > >
> > > We thank the reviewer for putting in the time to understand our work. We provide below an answer to their follow-up questions.
> > >
> > > 1. > _I think I understood your answer in that once you have a distilled dataset, learning a policy on top of it is similar to "offline" supervised training and thus you cannot really use another kind of policy learning algorithm for that task. Is that correct ?_
> > >
> > > That is correct. HaDES outputs a dataset of state-action pairs, and the only way we are aware of to obtain a policy from the dataset is by training with supervised learning (often called "behaviour cloning" in the context of RL).
> > >
> > > An alternative would be to train a dataset of state, action, reward and next state $(s, a, r, s')$ and do offline reinforcement learning [1]. However, we anticipate that there are many technical details to make such an approach work, and so we would consider this an entirely different method (i.e. not HaDES).
> > >
> > > 2. > _I understand now why it is necessary to use evolutionary strategies to avoid unfeasible computation of meta-gradients on a big number of inner steps. This is a good idea._
> > >
> > > Thank you!
> > >
> > > 3. > _As I am not familiar with these strategies, I am wondering however how well they can work when the state space gets bigger (in which case approximating the gradient with noise seems a more difficult problem). What is the typical "biggest" state space size that you have considered in your experiments ?_
> > >
> > > That is an excellent question. There is indeed evidence that ES tends to scale poorly with the dimensionality of the optimization problem (See section 3.2 of [2], with a theoretical result in [3]). However, [2] distinguishes between the number of parameters and the "difficulty, or intrinsic dimension, of the optimization problem".
> > >
> > > For Brax, the biggest state space is Humanoid, with size `|s|=244`. For MinAtar, the biggest state-space is Freeway (size `700`), followed by SpaceInvaders (size `600`).
> > >
> > > > _Could it for instance work on a state that is compounded of the features extracted from a resnet, of size 512_
> > >
> > > We expect the answer to yes, especially that those features would typically be continuous. In fact, performing behaviour distillation in some latent space is in our opinion one of the most promising ways to scale HaDES to even more complex settings than those in the paper.
> > >
> > > 4. > _In my previous questions, I was referring incorrectly sometimes to "dataset" and saying the HADES algorithm requires to access a "dataset" because I thought you were collecting a buffer of state-action pairs that you would then sample from as done in some algorithms. From your answer I understood that it is not the case, you learn this dataset online from episodes, is that correct ?_
> > >
> > > Correct. We learn a synthetic dataset that optimizes the return of the policy trained with supervised learning on that dataset. We discard the environment transitions and only use $J$ from the inner loop.
> > >
> > > Collecting a buffer of state-action pairs and using them to somehow improve the distillation process would be very interesting. However, it is not immediately clear to us how to do this.
> > >
> > > 5. > _About the comment on training speed, you can ignore it, I thought that these curves were comparing a method that is training on normal episodes vs methods that are training on distilled dataset and I was surprised to see no speed improvement. As I understand now, HADES is learning both a dataset and a policy, so these curves are just comparing HADES to ES and it makes sense that there is no speed improvement for the dataset learning part. I understand now that only Figure 4 looks at the results of training on top of HADES dataset._
> > >
> > > That is correct. We are glad we could clear up the misunderstanding.
> > >
> > > 6. > _After all of this, I think that the Figure 4 is the most important figure since it really shows the capacity of the distilled dataset. In my opinion it would be better to highlight this and maybe show it for different kind of environments instead of just Hopper._
> > >
> > > That is a good point. Figure 3 is about neuroevolution, whereas Figure 4 is really about behaviour distillation. For the final version of the paper, we will make sure to emphasise this result. For the time being, we show the equivalent plot for other environments in Section C.2 of the appendix.
> > >
> > > > _[HaDES] should be compared to more policy-learning algorithms additionally to ES._
> > >
> > > We have updated Figure 3 to include PPO, a very popular RL algorithm [4] and strong baseline for our settings [5].
> > >
> > > We hope this covers the reviewer's questions, and thank them once again for engaging in the discussion.
> > >
> > >
> > > [1] Survey on Offline RL, https://arxiv.org/pdf/2203.01387.pdf
> > >
> > > [2] Understanding Plasticity in Neural Networks, https://arxiv.org/pdf/2303.01486.pdf
> > >
> > > [3] Evolution Strategies as a Scalable Alternative to Reinforcement Learning, https://arxiv.org/pdf/1703.03864.pdf
> > >
> > > [4] PPO, https://arxiv.org/abs/1707.06347
> > >
> > > [5] ReLU to the rescue, https://arxiv.org/abs/2306.01460

---

> > > > ### Comment · Reviewer_ZN7t · 2023-11-20
> > > > **Score revision**
> > > >
> > > > Thanks for answering my questions. You have now cleared my doubts and I updated the score accordingly.

---

### Official Review · Reviewer_7Zz2 · 2023-10-29

**Soundness:** 3 good
**Presentation:** 3 good
**Contribution:** 2 fair
**Rating:** 6
**Confidence:** 4

**Summary:**

This manuscript proposed a behavior distillation algorithm of HaDES that aims to distill a few synthetic (state, action) pairs ($\mathcal{D}_\phi$) in the reinforcement learning (RL) setting, and the network can fastly learn a satisfied policy by training on the small distilled dataset. Concretely, the authors firstly formulate that behavior distillation problems as a bi-level optimization where the inner loop optimizes the policy network parameters by supervised learning on distilled data, while the outer loop maximizes the cumulative reward ($J$) w.r.t. the RL task.  Then, synthetic (state, action) pairs are optimized by gradient ascent where  the gradient is estimated by evolution strategies (ES). The authors verified that distilled dataset outperform vanilla ES algorithm on multiple RL datasets including Brax and MinAtar exvironments.

**Strengths:**

This paper first introduces the dataset distillation into the RL setting, and the core pros of this paper are presented as below:

1. Formulate the behaviour distillation by incorporating dataset distillation with an RL reward function;

2. Propose a novel behaviour distillation algorithm of HaDES that learns a few (state, action) pairs to fastly train a policy network with supervised learning;

3. Multiple empirical studies are conducted to verify the effectiveness and robustness of the synthetic behaviour dataset.

**Weaknesses:**

While the authors creatively incorporate dataset distillation into RL setting, there are some weaknesses mainly lie on the motivation and experiments, which hurt the contribution of this manuscript.

**Q1:** Sec. 1 states that this work is "motivated by the challenge of behaviour distillation", while this (challenge) is not a clear motivation for developing the behaviour distillation. What is the advantages of using the distilled dataset in RL except for fast training? In my opinion, the networks in RL are often small and do not require long time training.

**Q2:** While I am not an expert in RL, I note that the authors employ different networks to run HaDES and the vanilla ES baseline due to "memory constraints" so that HaDES outperforms vanilla ES, while synthetic datasets often largely underperform the real data in dataset distillation. It will be more reasonable to compare HaDES and vanilla ES under the same experimental settings. Moreover, it will make this work more convincing to add the results of other RL methods instead of only ES.

**Q3**: There lack of experiments investigating the influence of the distill budget (the size of the distlled dataset) on the final performance.

**Q4**: It will make this work more comprehensive if there is an analysis of efficiency. In detail, the author can list the GPU time of behaviour distillation, training on distilled dataset, vanilla ES and other RL methods, which can further highlight the importance of behaviour distillation.

Based on these observations, I think this manuscript is marginally below the acceptance, but I would like to increase my rating if the above questions are well addressed. Below are some minor questions and typos.

---

Minors and typos:
1. The term Behavior Distillation has already been used in [1], which is borrowed from the concept of knowledge distillation instead of dataset distillation. The authors should discuss this to avoid ambiguity.

2. The algorithm 1 should be placed in the main text for clear illustration.

2. Page 2: "scaling independently" -> "scaling independence"

3. Page 2: "policy, reducing" -> "policy, thereby reducing"

4. Page 3: "train a model faster" -> "train a model fastly"

5. Page 4: The formulation of dataset $\mathcal{D}$: "$\mathcal{D} = \\{x_i, y_i\\}$" -> "$\mathcal{D} = \\{x_i, y_i\\}_{i=1}^N$

6. Page 5: this first line: ", i.e. " -> ", i.e., "

7. Page 5: there exists an extra right bracket in Eq. (3)

[1] Furuta, Hiroki, et al. "A System for Morphology-Task Generalization via Unified Representation and Behavior Distillation." The Eleventh International Conference on Learning Representations. 2022.

**Questions:**

See weaknesses.

---

> ### Author Response · Authors · 2023-11-17
> **Response to R2**
>
> We thank Reviewer 7Zz2 for their valuable comments. In particular, we welcome the helpful suggestions on how to improve our experiments section. We invite the reviewer to read our response below.
>
> 1. > _"In my opinion, the networks in RL are often small and do not require long time training."_
>
> We agree that the networks used in RL tend to be smaller than the convolutional networks used in vision. However, RL regularly requires tens of millions of environment steps to learn a good policy, which can easily translate to several hours or even days of training time [1, 2]. This is why there has been a recent push to build hardware-accelerated RL environments and high-performance RL implementations [2,3,4,5,6]. The distilled datasets allow us to completely avoid environment bottlenecks by turning policy training into a very small supervised learning problem.
>
> 2. > _"advantages of using the distilled dataset in RL except for fast training?"_
>
> Beyond training speed, there are multiple applications to the datasets produced by HaDES. Notably:
>
> - In section 5.2, we show that distilled datasets readily generalise to a large range of architectures and hyperparameters, which implies they can be used for effective Neural Architecture Search (NAS). NAS has notably been identified as a potential application of Dataset Distillation in the supervised domain as well [7].
>
> - In section 5.3, we demonstrate that the datasets evolved for individual environments can be combined to train a multi-task agent, therefore enabling faster research on generalist foundational models.
>
> - In section 5.5, we show briefly that the datasets can help explain the behaviour of the policies and make them more interpretable.
>
> - As mentioned in our conclusion, we also hope HaDES to act as a starting point for new and improved methods for continual learning, where resetting and retraining policies repeatedly is common to fight plasticity loss [2, 9].
>
> 3. > _"It will be more reasonable to compare HaDES and vanilla ES under the same experimental settings"_
>
> Thank you for raising this point. We have now updated Figure 3 b) with additional runs where the network widths match those used for vanilla ES (width=256). The impact of the on the final performance is negligible in SpaceInvaders, Asterix and Freeway, but has a noticeable impact on Breakout. Nonetheless, the order of the different methods remains unchanged.
>
> For Brax (Figure 3.a)), this was already the case (both HaDES and ES use `width=512`). Nonetheless, HaDES-F outperforms ES in all Brax environments tested, and HaDES-R matches or beats the baseline in 6/8 environments.
>
> 4. > _"add the results of other RL methods instead of only ES"_
>
> We agree that reporting an RL baseline in addition to ES would help contextualise our results better. We have updated Figure 3 with a PPO baseline, which is a very strong RL baseline for our suite of environments [10]. While ES methods still lag behind RL in terms of performance, HaDES helps narrow the gap significantly. This is in addition to the main benefit of HaDES, namely producing the distilled datasets.
>
> 5. > _"lack of experiments investigating the influence of the distill budget (the size of the distilled dataset) on the final performance"_
>
> We thank the reviewer for this helpful suggestion. We have added such an analysis to section C.1 of the appendix, with a reference to it at the beginning of section 5.
>
> 6. > _"list the GPU time of behaviour distillation, training ..."_
>
> We have now expanded section B.2 with additional details regarding the GPU time for different methods. In particular, please refer to Table 4 for a breakdown per environment.
>
> 7. > Minor points:
>
> We thank the reviewer for their thoroughness! We have added a clarification regarding the term "Behaviour Distillation" to avoid ambiguity. Regarding the algorithm, we have considered moving it from the appendix to the main paper, but have decided against it due to lack of space.
>
> We hope that we addressed the reviewer's comments, which genuinely helped us improve our manuscript! As a result, we hope the reviewer will consider recommending our revised submission for acceptance.
>
> [1] R2D2, https://openreview.net/pdf?id=r1lyTjAqYX
>
> [2] Bigger, Better, Faster, https://arxiv.org/abs/2305.19452
>
> [3] Madrona, https://madrona-engine.github.io/shacklett_siggraph23.pdf
>
> [4] Brax, https://arxiv.org/abs/2106.13281
>
> [5] Scaling Distributed RL, https://arxiv.org/abs/2306.16688
>
> [6] MuJoCo 3 announcement, https://github.com/google-deepmind/mujoco/discussions/1101
>
> [7] Data Distillation: A Survey, https://arxiv.org/pdf/2301.04272.pdf
>
> [8] PureJaxRL, https://github.com/luchris429/purejaxrl
>
> [9] The Primacy Bias in Deep RL, https://arxiv.org/abs/2205.07802
>
> [10] ReLU to the rescue, https://arxiv.org/abs/2306.01460

---

> ### Comment · Reviewer_7Zz2 · 2023-11-19
>
> I appreciate the authors' response, which clears part of my concerns. I still have some concerns as below:
>
> **Q1:** "The distilled datasets allow us to completely avoid environment bottlenecks by turning policy training into a very small supervised learning problem." Does the vanilla policy training use supervised learning on an un-distilled large (state, action) pair dataset? If yes, I think the behavior distillation still lies in a supervised learning regime instead of reinforcement learning.
>
>
>
> **Q2:** The authors show many pros of behavior distillation in NAS, interpretability, continual learning, etc. In my opinion, these pros are also inherent advantages for dataset distillation and lie in supervised learning regime. How about the exclusive advantage of behavior distillation *especially in RL field*.
>
>
>
> **Q3:** There still lack experiments on `ES neuroevolution, width=512` in Fig. 3(b)
>
>
>
> **Q4:** I am curious why networks trained on small distilled data can outperform those trained on large source real data in behavior distillation, while the generalization performance of distilled largely lags behind the real data in dataset distillation.
>
> **Q5:** Can the behavior distillation integrate with other RL algorithms instead of ES? If ES largely lags to modern mainstream RL algorithms, the contribution of behavior distillation will be greatly undermined.
>
>
>
> **Q6:** Because efficiency is a main advantage for behavior distillation, a runtime analysis in necessary. However, according to results in Table 4, (1) ES has shorter runtime than HaDES, which means ES is more efficient, am I right? (2) The time unit in Table 4 is second, which means that both training with vanilla setting or behavior distillation have a satisfied speed and thus undermines the necessity to develop behavior distillation. This is why I asked Q1 in my (initial) official review.

---

> > ### Author Response · Authors · 2023-11-19
> > **2nd Response to R2**
> >
> > We thank the reviewer for engaging further in the discussion and hope our answer will clear the last of their concerns.
> >
> > **A1:**
> >
> > No, vanilla policy training does not use supervised learning. We report two baselines: PPO, which is a strong and very popular RL algorithm [1], and ES neuroevolution (Figure 2, left), which simply evolves the policy parameters to maximise $J$. Neither use a real dataset; they learn from the environment rewards. The same holds for our method.
> >
> > To be clear, at **no** point do we do supervised learning on real state-action pairs.
> >
> > **A2:**
> >
> > We wholeheartedly agree that many of the pros of BD are inherited from DD. Extending the benefits of DD to RL was one of the main motivations of this paper.
> >
> > However, as explained in Section 1, BD provides a unique advantage to RL, which is to turn the RL problem into a supervised one. This eliminates the need for exploration [7] or for handling non-stationarity [2] -- two challenges that are specific to RL. In practice, this means for instance that we can decouple NAS from algorithmic decisions (e.g. which RL algorithm to use). It also allows to train policy networks without learning a critic [8], eliminating a lot of the complexity of policy-based methods.
> >
> > **A3:**
> >
> > Those experiments are not missing; we are unable to run ES with `population_size=2048` and `width=512` on our hardware (8x Tesla V100-SXM2-32GB). As we explain in Section 5.1, we get an Out Of Memory error because the entire population must be located on a single GPU at once before running each inner loop. We tried ES with `(population_size=1024, width=512)` and `(population_size=2048, width=256)`, and reported the latter because it performs better.
> >
> > In contrast, HaDES generates a population of datasets, which have much fewer parameters than policy networks. It then distributes the datasets and runs the inner loops on multiple GPUs. The policies are initialised in the inner loops and are never colocated on a single GPU, so HaDES has a much lower peak memory usage. This means we can run bigger populations and bigger networks using the same hardware.
> >
> > **A4:**
> >
> > This is indeed interesting, and explaining it fully would require an in-depth analysis of learning dynamics that goes beyond the scope of the paper. However, we hyposize two contributing factors:
> > 1. ES tends to scale poorly with the number of parameters being optimized (See section 3.2 of [3], and theoretical result in [4]). By evolving datasets rather than networks, HaDES optimises a much smaller number of parameters. For instance, in `halfcheetah`, each network has $396834$ parameters, while the dataset of size $64$ has only $16704$ parameters.
> > 2. HaDES performs optimization in dataset-space, not policy-space. It is possible that the optimization landscape in the dataset-space is easier, or leads to solutions not easily discovered by RL, e.g. the exploit that HaDES discovers in `humanoid-standup`.
> >
> > **A5:**
> >
> > BDn can in theory be done with approaches that do not rely on ES, e.g. by solving the outer loop in eq. 3 with meta-gradients or any other optimization method, or doing Offline Reinforcement Learning [5] in the inner loop
> >
> > We merely propose one approach, HaDES, which solves the bi-level optimization in section 4.1 directly using ES in the outer loop and Behaviour Cloning in the inner loop.
> >
> > **A6:**
> >
> > Looking at Table 4, performing BD using HaDES is indeed slower than training a policy with ES. However, once we are done, the time required to train a policy using the final distilled dataset is equal to just 1 HaDES generation. In short:
> > - Avg. time to train 1 policy with ES: `num_gen * 4.46s` (8 GPUs)
> > - Avg. time to perform behaviour distillation with HaDES-R: `num_gen * 7.98s` (8 GPUs)
> > - Avg. time to train 1 policy on the final dataset:  `7.98s` (1 GPU)
> >
> > Consider `halfcheetah`. ES took 2000gen * 10.6s/gen = 5.94h to train 1 policy. HaDES-R took 2000gen * 16.9s/gen = 9.39h to do BD and produce a synthetic dataset. But then training additional policies using the synthetic dataset takes only as much as a single HaDES-R generation, i.e. 16.9s. This represents a **1266x speedup** with respect to ES.
> >
> > This also holds for dataset distillation: training a classifier on MNIST or CIFAR-10 data takes a few minutes. Doing dataset distillation requires 1 to 4 hours on 1 to 4 V100 GPUs [6], but training a classifier on the distilled dataset takes seconds.
> >
> > [1] PPO Algorithms, https://arxiv.org/abs/1707.06347
> >
> > [2] Understanding Plasticity, https://arxiv.org/pdf/2303.01486.pdf
> >
> > [3] ES as a Scalable Alternative to RL, https://arxiv.org/pdf/1703.03864.pdf
> >
> > [4] Random Gradient-Free Minimization of Convex Functions, https://link.springer.com/article/10.1007/s10208-015-9296-2
> >
> > [5] Survey on Offline RL, https://arxiv.org/pdf/2203.01387.pdf
> >
> > [6] DD, https://arxiv.org/pdf/1811.10959.pdf
> >
> > [7] Survey of Exploration, https://arxiv.org/abs/2109.00157
> >
> > [8] Survey of actor-critic RL, https://hal.science/hal-00756747/file/ivo_smcc12_survey.pdf

---

> > > ### Comment · Reviewer_7Zz2 · 2023-11-20
> > >
> > > Thanks for the author's response, which clears all my concerns. I will correspondingly raise my rating.
> > >
> > > One minor: I suggest the authors move the analysis of efficiency (GPU time, distillation budget) to the main text to better demonstrate the superiority of the proposed HaDES.

---

### Official Review · Reviewer_f5UH · 2023-10-31

**Soundness:** 3 good
**Presentation:** 3 good
**Contribution:** 3 good
**Rating:** 6
**Confidence:** 3

**Summary:**

This paper proposes a new branch of dataset distillation - behaviour distillation. Behaviour distillation aims at distilling a set of synthetic RL dataset without having access to the expert data. The authors handles indifferentiability of the formulation using evolution strategy, and show good performance.

**Strengths:**

+ The introduction of behaviour distillation can enrich the literature and direction of Dataset Distillation. Different from standard DD, behaviour distillation does not require access to the expert datasets. This is quite close to a standard /basic RL setting and could be a good starting point.

+ The author's writing is easy to follow and pretty clear on the technical details

+ The experimental section show promising results using the proposed algorithm HADES.

**Weaknesses:**

- Although it is interesting, the proposed behaviour distillation seems to not have a clear motivation on why it can be useful (what's the motivation for proposing this problem and it's potential application, besides DD hasn't been applied to RL), and why not directly formulating the problem on expert dataset. Distilling directly from scratch can make the problem a lot harder.
- It would be great if the authors can discuss the behaviour difference of a standard RL algorithm and a synthetic data-driven RL algorithm (HADES). What's the potential benefits?
- One missing citation on BPTT [2]. It would be nice to add some discussion on optimization algorithm in DD, for example BPTT (the original one[1] and momentum-based[2]).

Overall the paper is quite interesting. Looking forward to authors' response on the above questions.

[1] Dataset Distillation

[2] Remember the Past: Distilling Datasets into Addressable Memories for Neural Networks

**Questions:**

See above.

---

> ### Author Response · Authors · 2023-11-17
> **Author Response to R1**
>
> We thank reviewer f5UH for their feedback and are glad they believe our paper can enrich the literature surrounding Dataset Distillation.
>
> 1. Behaviour Distillation as a setting is directly motivated by the success of Dataset Distillation in the supervised domain, with applications such as differential privacy, neural architecture search, continual learning, and federated learning [1]. With our work, we hope to extend the reach of those applications to RL, while also enabling new RL-specific ones, such as the training of multi-task policies without additional environment interactions (section 5.3 of our paper).
>
> We formulated Behaviour Distillation to be independent of an expert policy for two main reasons:
> - Firstly, an expert dataset is not always available, and so we do not wish to rely on this assumption. Furthermore, even if an expert dataset was available, the expert's behaviour could be erratic and hard to distill, which means that we would need to increase the size of the dataset to capture the idiosyncrasies of the expert. Instead, HaDES restricts its optimization to policies that can be distilled into datasets of size $N$, eliminating policies that are too erratic by design.
> - Secondly, even when we do have access to a dataset, supervised dataset distillation is still likely to fail. In RL, a policy trained with behaviour cloning (BC) on an expert dataset incurs regret with respect to the expert that is quadratic in the length of the episode [2]. Intuitively, this happens because each timestep $\pi$ deviates from $\pi_{expert}$, it risks ending up off-distribution, which in turn makes it more likely to make additional mistakes. Since dataset distillation is often lossy, we can therefore expect $\pi$ to achieve low returns in many long horizon tasks, even if it achieves low (but non-zero) cross-entropy with the expert.
>
> We recognize this motivation wasn't explicit in the original manuscript and have therefore expanded section 4.1 with the details above.
>
> 2. > _"discuss the behaviour difference of a standard RL algorithm and a synthetic data-driven RL algorithm (HADES). What's the potential benefits?"_
>
> We are not sure whether the reviewer is asking about the difference between algorithms or between the resulting policies. We answer both:
>
> - Differences between algorithms:
>   - Deep RL algorithms, in short, will repeatedly unroll the policy in the environment to collect transitions, and use those transitions to update the policy through SGD, in a way that ultimately maximises the expected discounted return.
>   - Vanilla Evolutionary Strategies (ES) perform Neuroevolution: they start from a randomly initialised neural network parametrization $\theta$ of the policy $\pi_\theta$. Each generation, they sample a population of random variations of that parametrization, estimate the expected return of each member of the population (the candidates), and use that information to update the policy parameters.
>   - As described in section 4.2, HaDES is similar to ES, but instead of keeping track of a policy $\pi_\theta$, it keeps track of a synthetic dataset $D_\phi$. Each generation, it then samples a population of datasets and uses BC to obtain a policy from each dataset candidate. The advantage is that $|\phi|$ < $|\theta|$ typically, which reduces the single-GPU memory of the algorithm and means there are fewer parameters to optimise. In practice, this also translates to significantly higher returns, as shown in Section 5.1. Additionally, the advantage of HaDES over both Deep RL and ES is that it produces the dataset $D_\phi$, which can be used for fast retraining and other applications.
>
> - Differences between policies:
>   - Because HaDES and Deep RL rely on very different optimization methods, they can in theory produce policies with different behaviours. For instance, HaDES only restricts optimization over relatively "simple" policies, i.e. those that can be condensed in a dataset of predetermined size. This is beneficial in scenarios where interpretability and predictability are important. We believe exploring the full impact of the training method on the resulting policies would be a promising direction for future work.
>
> 3. > _"One missing citation on BPTT."_
>
> Thank you for the missing reference. We have now added it to the paper.
>
> 4. > _"It would be nice to add some discussion on optimization algorithm in DD, for example BPTT"_
>
> We already had a brief discussion regarding the connection between BPTT/meta-gradients and HaDES in section 2.1. Following feedback from the reviewer, we have added a few more details to the revised manuscript. If this is not what the reviewer meant, we ask them to kindly clarify.
>
> We hope this covers the reviewer's questions, and are happy to continue the conversation should the reviewer have any follow-up.
>
> [1] Data Distillation: A Survey, https://arxiv.org/pdf/2301.04272.pdf
> [2] Efficient Reductions for Imitation Learning, https://proceedings.mlr.press/v9/ross10a.html

---

> > ### Author Response · Authors · 2023-11-22
> > **Follow-up from the authors**
> >
> > We hope our response has addressed the reviewer's concerns. In addition, we have uploaded a revised manuscript with the following improvements:
> >
> > 1. Added the extra citation on BPTT and clarified our choice of using ES rather than BPTT. (Section 2.1)
> > 2. Clarified our motivation and especially why we formulate the problem without access to expert data (Section 4.1)
> > 3. Added PPO as a baseline for both Brax and MinAtar (Figure 3)
> > 4. Added results for HaDES with a reduced network size in MinAtar (Figure 3 b))
> > 5. Added per-environment wall clock time comparisons between ES and HaDES (Appendix B.2)
> > 6. Added experiments investigating the effect of the distillation budget on the policy return (Appendix C.1)
> >
> > If the reviewer has additional feedback, we kindly ask them to share so that we may further improve the paper. Otherwise, we hope they will consider increasing their support to our submission.

---

### Meta-Review · Area_Chair_DTrq · 2023-12-11

**Metareview:**

The paper introduces an innovative concept in dataset distillation, termed "behavior distillation," which focuses on distilling a synthetic dataset for reinforcement learning (RL) without the need for expert data. The reviewers have collectively expressed appreciation for the results demonstrated in the study, showing a general inclination towards accepting the paper. However, the area chair has some reservations, particularly regarding the simplicity of the test environments used in the research. There is a concern about whether the proposed method would be equally effective in more complex scenarios, such as object manipulation environments that involve intricate contact dynamics. Despite these reservations, the area chair acknowledges the novelty of this being the first work on behavior distillation. This recognition of its pioneering nature leads to a tendency to support exploratory work in this new area. Consequently, the area chair is inclined to endorse the paper, encouraging further exploration and development in the field of behavior distillation.

**Justification For Why Not Higher Score:**

The method may not be effective for object manipulation tasks which are more practically than those experimented in the work.

**Justification For Why Not Lower Score:**

All reviewers are positive of the work.

---

### Decision · Program_Chairs · 2024-01-16

Accept (poster)